# Identification of Elite *R*-Gene Combinations against Blast Disease in *Geng* Rice Varieties

**DOI:** 10.3390/ijms24043984

**Published:** 2023-02-16

**Authors:** Peng Gao, Mingyou Li, Xiaoqiu Wang, Zhiwen Xu, Keting Wu, Quanyi Sun, Haibo Du, Muhammad Usama Younas, Yi Zhang, Zhiming Feng, Keming Hu, Zongxiang Chen, Shimin Zuo

**Affiliations:** 1Key Laboratory of Plant Functional Genomics of the Ministry of Education/Jiangsu Key Laboratory of Crop Genomics and Molecular Breeding, Agricultural College of Yangzhou University, Yangzhou 225009, China; 2Co-Innovation Center for Modern Production Technology of Grain Crops of Jiangsu Province/Key Laboratory of Crop Genetics and Physiology of Jiangsu Province, Yangzhou University, Yangzhou 225009, China; 3Joint International Research Laboratory of Agriculture and Agri-Product Safety, The Ministry of Education of China, Institutes of Agricultural Science and Technology Development, Yangzhou University, Yangzhou 225009, China

**Keywords:** *Geng* rice cultivars in Jiangsu, blast resistance, *R* genes, gene combinations

## Abstract

Rice blast, caused by the *Magnaporthe oryzae* fungus, is one of the most devastating rice diseases worldwide. Developing resistant varieties by pyramiding different blast resistance (*R*) genes is an effective approach to control the disease. However, due to complex interactions among *R* genes and crop genetic backgrounds, different *R*-gene combinations may have varying effects on resistance. Here, we report the identification of two core *R*-gene combinations that will benefit the improvement of *Geng* (*Japonica*) rice blast resistance. We first evaluated 68 *Geng* rice cultivars at seedling stage by challenging with 58 *M. oryzae* isolates. To evaluate panicle blast resistance, we inoculated 190 *Geng* rice cultivars at boosting stage with five groups of mixed conidial suspensions (MCSs), with each containing 5–6 isolates. More than 60% cultivars displayed moderate or lower levels of susceptibility to panicle blast against the five MCSs. Most cultivars contained two to six *R* genes detected by the functional markers corresponding to 18 known *R* genes. Through multinomial logistics regression analysis, we found that *Pi-zt*, *Pita*, *Pi3/5/I*, and *Pikh* loci contributed significantly to seedling blast resistance, and *Pita*, *Pi3/5/i*, *Pia*, and *Pit* contributed significantly to panicle blast resistance. For gene combinations, *Pita*+*Pi3/5/i* and *Pita*+*Pia* yielded more stable pyramiding effects on panicle blast resistance against all five MCSs and were designated as core *R*-gene combinations. Up to 51.6% *Geng* cultivars in the Jiangsu area contained *Pita*, but less than 30% harbored either *Pia* or *Pi3/5/i*, leading to less cultivars containing *Pita*+*Pia* (15.8%) or *Pita*+*Pi3/5/i* (5.8%). Only a few varieties simultaneously contained *Pia* and *Pi3/5/i*, implying the opportunity to use hybrid breeding procedures to efficiently generate varieties with either *Pita*+*Pia* or *Pita*+*Pi3/5/i*. This study provides valuable information for breeders to develop *Geng* rice cultivars with high resistance to blast, especially panicle blast.

## 1. Introduction

Rice blast, a fungal disease caused by *Magnaporthe oryzae*, is one of the top three rice diseases worldwide that seriously threatens rice production each year, causing 10–30% yield reductions in popular years and up to 50% loss in severe cases [1,2]. Rice blast occurs throughout the entire reproductive period of rice and thus can be divided into seedling blast, leaf blast, node blast, panicle blast, and grain blast depending on the period of infestation and the location of the disease on rice plant. Among them, panicle blast poses the greatest threat to rice production [3]. Unlike usage of chemical fungicides, which have many disadvantages in controlling blast disease [4], the selection and rational application of resistant rice varieties is the most economical and effective approach to the control of the disease [5]. Consequently, the identification of rice blast resistance (*R*) genes has become a prerequisite for breeding resistant cultivars.

To date, more than 100 qualitative *R* genes to rice blast have been reported [6], and 38 of them have been successfully cloned from different rice varieties [7], providing effective genetic resources for rice blast resistance breeding. However, due to the high variability of *M. oryzae* populations and the emergence of new virulent strains, many resistant varieties that contain the individual *R* gene may lose resistance after a period of large-scale cultivation [8]. Therefore, gene pyramiding using different *R* genes has been considered a more effective approach to preventing the loss of resistance in varieties and to breeding varieties with durable, broad-spectrum resistance [9]. For example, Wu et al. (2019) used molecular-marker-assisted selection (MAS) to pyramid *Pigm* and *Pi1*, *Pi54*, and *Pi33*, and the pyramiding lines showed stable broad-spectrum resistance under artificial inoculation at both seedling and booting stages [1]. Xiao et al. (2016) introduced *Pi46* and *Pita* into the superior restorer Hanghui 179 line via MAS, finding that all three lines showed significant improvement on blast resistance [10]. However, in these studies, they also found that different *R*-gene combinations showed clearly different pyramiding effects on resistance, suggesting that gene interactions may widely exist among different *R* genes [1,11]. Moreover, it was noted that some *R* genes or pyramiding of many *R* genes affected plant development and grain yield [1,12]. Therefore, it is particularly important to identify appropriate *R*-gene combinations that stably express positive interactions or additive effects on resistance without inferior effects on grain yield in diversified genetic backgrounds of rice cultivars.

Jiangsu Province is one of the major rice production areas in China, where more than 85% of *Geng* rice varieties are grown, accounting for 20% of the total area and 30% of the total output of *Geng* rice in China [2,13]. In recent years, with the change of farming system and climatic conditions, *Geng* rice production in Jiangsu Province has been increasingly affected by rice blast disease. In order to understand *R* genes utilized in rice varieties in this area and to better guide subsequent development of resistant varieties, several studies have investigated the distribution of some known *R* genes by using functional or diagnostic markers [14,15,16,17,18,19]. In these studies, they found that *Pita*, *Pib*, *Pikm*, *Pi54*, and *Pi-zt* possess good application prospects in Jiangsu for *Geng* rice breeding against blast [16,17,18,19]. Moreover, they found that *Pib*, *Pita*, *Pikh*, *Pb1*, *Pid2*, *Pid3*, *Pikm*, and *Pik* have been widely utilized in Jiangsu *Geng* rice varieties with distribution frequencies ranging from 20% to 70%, while *Pi2*, *Pi9*, and *Pit* were present in very few varieties [16] and *Pigm* existed in almost no cultivars. Most recently, increasing lines of evidence showed that the *Pigm* gene has a great potential in developing *Geng* rice cultivars with durable, broad-spectrum resistance against blast in Jiangsu province [15]. Two new *Geng* cultivars, Yangnognjing 3091 and Jingxiangyu 1, were developed with *Pigm* through MAS that showed broad-spectrum resistance at seedling stage as well as excellent performance on panicle blast resistance to more than 184 isolates collected from the Yangtze River area [12,15]. Besides this individual gene with such an excellent effect on blast resistance, Wang et al. (2020) found that two *R*-gene combinations, *Pita*+*Pia* and *Pita*+*Pi3/5/i*, had great potentials for developing *Geng* rice cultivars with panicle blast resistance in Jiangsu Province. Notably, they found that while individual *Pita* gene had almost no effects on panicle blast resistance, it significantly increased resistance effects when pyramided with *Pia* or *Pi3/5/i*, suggesting that positive interactions or additive effects exist between the two genes [14]. However, due to a relatively small number of *M. oryzae* isolates used in the study, this interesting conclusion remains to be further investigated.

In this study, we employed a total of 190 elite *Geng* rice cultivars/lines developed in recent years in Jiangsu Province to identify known *R* genes or *R*-gene combinations that benefit the further improvement of *Geng* rice resistance against blast in breeding practice. Twenty-nine *M. oryzae* isolates collected from different rice growing regions in Jiangsu Province were used in five groups of mixed conidial suspensions (MCSs), with each MCS containing 5–6 isolates to inoculate artificially for evaluating panicle blast resistance. A total of 18 *R* gene loci, including the *Pi2/9* locus, were detected by using functional or diagnostic markers of each gene. Through multinomial logistics regression analysis, we identified several *R*-gene combinations that positively enhance rice resistance to different MCSs in panicle blast inoculations; among them, *Pita*+*Pi3/5/i* and *Pita*+*Pia* were designated as elite core *R*-gene combinations. However, we found that no more than 20% cultivars harbored either of the two core *R*-gene combinations, and almost no cultivars simultaneously contain *Pia*+*Pi3/5/i*, which affected the efficiency to rapidly develop new cultivars with these core *R*-gene combinations. Together, this study not only reveals valuable information for *Geng* rice breeding against blast, especially for panicle blast, in the Jiangsu area, but also provides a reference for other ecoregions to identify elite *R* genes or *R*-gene combinations for guiding blast resistance breeding.

## 2. Results

### 2.1. Resistance Evaluation of Geng Rice Cultivars to Blast Disease

In order to evaluate the *Geng* rice varieties in the Jiangsu area for rice blast resistance, we artificially inoculated 68 varieties at seedling stage with 58 *M. oryzae* isolates collected from 11 cities in the Jiangsu area (Figure 1A). We found that 28 of the 68 varieties (41.2%) had a resistance frequency (RF) for more than 90% isolates (Figure 1B). In total, more than 80% varieties showed resistance to more than 80% isolates, suggesting that Jiangsu *Geng* rice varieties have preferable resistance at seedling stage. Further, we selected 24 *M. oryzae* isolates on the basis of their geographic locations and different genotypes on 10 *avirulence* gene loci detected by the corresponding functional markers (Appendix A). After cultivation, these isolates were then randomly mixed into four groups (MCS2-MCS5), with each group containing six isolates. MCS2-MCS5, together with MCS1, consisting of five representative isolates from Yongfeng Liu’s lab, were used to inoculate 190 *Geng* rice varieties at the booting stage. As shown in Figure 1C, the proportions of varieties with disease grades in the range of seven to nine inoculated with the five MCSs were 69%, 62.2%, 66.5%, 70.2%, and 78.7%, respectively, indicating that most varieties were susceptible or highly susceptible to panicle blast. Positive correlations between the disease grades of varieties and the MCSs were detected though their correlation coefficients and were not very high, ranging from 0.344 to 0.694 (Figure 1D). This indicates that the five MCSs represent a certain degree of diversity, which is very valuable for identifying elite *R* genes or *R*-gene combinations against diverse *M. orzyae* isolates in the field.

### 2.2. Analysis of Known R Genes and Their Combinations for Seedling Blast Resistance

In order to explore known *R* genes and *R*-gene combinations that contribute significantly to rice blast resistance, we firstly detected the genotypes of 18 known *R* gene loci for all 190 *Geng* rice varieties using their corresponding functional molecular markers (Appendix A). Although five *R* genes in the Pi2/9 locus were detected, we found that only *Pi-zt* was detected in around 30% varieties and that no varieties contained the other four genes except for one variety containing *Pigm* (Figure 2A). The remaining 13 *R* genes showed a varying distribution ranging from 3.2% to 81.1% in these rice varieties, with *Pib* having the highest distribution frequency at 81.1% and *Pi25* the lowest frequency at 3.2% (Figure 2A). Most varieties contained 2–6 tested *R* genes, while only three varieties contained 8–9 *R* genes (Figure 2B). Moreover, we found that the panicle blast disease grades in each MCS inoculation were significantly negatively correlated with the number of *R* genes in the varieties with a coefficient of 0.0356–0.2108 (*R*^2^), indicating that the varieties with more *R* genes generally have higher resistance and vice versa (Appendix A). 

Through multiple linear regression analysis using the phenotype of 68 varieties on seedling blast resistance, we found that four genes, *Pi-zt*, *Pita*, *Pi3/5/i* and *Pikh*, showed significant contributions to seedling blast resistance (Figure 2C). Furthermore, we compared the RF of varieties with different numbers of the four genes, not including varieties with target genes or gene combinations less than three. As shown in Figure 2D, the varieties without these four genes showed significantly lower RF, being 53.4% lower than the varieties with one or more of these genes. The highest RF was 94.4% found in varieties with *Pikh*+*Pi3/5/i*+*Pita*, and the lowest RF was 81% in varieties containing *Pikh*+*Pi3/5/i*; no statistically significant differences were found among the varieties with different combinations of the four genes. Because of the relatively high proportion of *Pita* and *Pikh* in these varieties, we speculate that these two genes may be the main factors accounting for the high resistance of these Jiangsu *Geng* rice varieties at seedling stage and that a further increase in the frequency of the other two *R* genes (*Pi-zt* and *Pi3/5/i*) may be an appropriate approach to further improving the seedling blast resistance of *Geng* varieties in Jiangsu area. 

### 2.3. Analysis of Elite R Genes and R-Gene Combinations for Panicle Blast Resistance

In order to identify elite *R* genes or *R*-gene combinations on panicle blast resistance, we performed a multinomial logistics regression analysis using the genotype data of the *R* genes tested and the disease grades of the varieties (Appendix A). Four *R* genes, *Pigm*, *Piz*, *Pi2*, and *Pi9*, were only detected in few varieties and therefore were not included in this analysis. Among the remaining 14 *R* genes, we found that different numbers of *R* genes were detected with significant contributions to resistance in inoculations with the five different MCSs (Appendix A). In total, eleven *R* genes with significant contributions were detected in different MCS inoculations and designated as elite *R* genes hereafter (Figure 3A). MCS2 inoculation detected three elite *R* genes, the lowest number, and MCS3 detected seven, the highest number. Although elite *R* genes were spread in different MCS inoculations, a few common elite *R* genes were observed. *Pita* and *Pi3/5/i* were found in all five MCS inoculations, and *Pit* and *Pia* were detected in four and three MCS inoculations, respectively. *Pid3* and *Pik* were detected in two different MCS inoculations, and *Pid2*, *Pi-zt*, *Pikh*, *Pi1*, and *Pb1* were only found in one MCS inoculation. Three *R* genes, *Pikm*, *Pib*, and *Pi25*, were found without significant contributions to panicle blast resistance in any of the five MCS inoculations. 

To determine the resistance contributions of these elite *R* genes against the corresponding MCS, we then performed a multiple comparison analysis using the disease grades of varieties containing different elite *R* genes and their combinations (Appendix A). Interestingly, under individual *R* gene status, most elite *R* genes did not confer significant effects on reducing disease grades. For gene combinations, we found that, except for very few *R*-gene combinations, most *R*-gene combinations significantly reduced disease grades compared to individual *R* genes. For example, in MCS1, four combinations, *Pita*+*Pid2*, *Pi3/5/i*+*Pita*, *Pia*+*Pita*, and *Pia*+*Pita*+*Pid2*, displayed significantly increased resistance; only the combination *Pia*+*Pid2* did not show significant difference on disease grades compared with the control without elite *R* genes (Appendix A). These results suggest that complex interactions do exist among different *R* genes. Although some elite *R* genes individually had no significant resistance effects, they were required for other *R* genes to confer resistance.

We thus further summarized the elite *R* genes and *R*-gene combinations that showed significantly higher resistance effects than the control for each MCS inoculation (Appendix A). Collectively, three elite *R* genes, *Pi3/5/i*, *Pit*, and *Pita*, and 15 *R*-gene combinations were identified, and some of them were commonly observed in different MCS inoculations (Figure 3B). For the three elite *R* genes, *Pi3/5/i* and *Pit* showed significant effects even as an individual gene in both MCS1 and MCS2 inoculations. Three double-gene combinations, *Pita*+*Pi3/5/i*, *Pita*+*Pia*, and *Pita*+*Pid3*, were commonly detected in four, three, and two inoculations with different MCSs, respectively. No common combinations with three or four genes were found. We further compared the average disease degrades of varieties containing each of the 3 elite *R* genes and the 15 *R*-gene combinations against all five MCSs (Figure 3C). We found that, except for two *R*-gene combinations (*Pikh*+*Pia*+*Pi-zt* and *Pik*+*Pita*), all *R*-gene combinations showed an average disease grade of 4 or lower. From the distribution of disease grades, we also observed that most varieties containing any of these 13 *R*-gene combinations had disease grades no more than 5, demonstrating their clear contributions and high breeding potentials. Taken together, these results suggest that 3 *R* genes and 13 *R*-gene combinations have potential for rice breeding against panicle blast, at least in the Jiangsu area.

### 2.4. Comparison of Varieties with Different R-Gene Combinations on Resistance at Both Seedling and Booting Stages

Because we obtained the data of disease resistance at both seedling and booting stages for 68 varieties, we then compared the resistance effects of varieties with *R*-gene combinations at both developmental stages. As shown in Figure 4A, we found that the panicle blast disease grade was significantly negatively correlated with the RF at seedling stage with a co-efficient of 0.3097 (*R*^2^), indicating that high RF varieties generally tend to have stronger resistance on panicle blast and vice versa. Subsequently, we analyzed the panicle blast resistance performance of the varieties with elite *R*-gene combinations identified from seedling stage tests (Figure 2D and Figure 4B). Except for the *Pikh*+*Pi-zt* combination, we found that varieties with each of the remaining five combinations all showed significantly lower panicle blast disease grades than their corresponding controls. Compared with the varieties without these elite *R* genes identified in seedling blast tests, all six *R*-gene combinations showed a significant reduction on panicle blast disease grades (Figure 4C). However, these gene combinations did not differ significantly on their seedling stage RF data but showed differences on panicle blast disease grades (Figure 2D and Figure 4C). 

Similarly, the seedling blast resistance performance of the 68 varieties with different elite *R*-gene combinations identified in panicle blast tests was also analyzed (Figure 3B and Figure 4D). The results showed that the varieties with each of the nine *R*-gene combinations all had significantly higher RF than their corresponding controls (Figure 3B and Figure 4D). In particular, eight of nine combinations had RF values higher than 90%, while no significant differences were found among one another (Figure 3C and Figure 4E). Only two *R*-gene combinations (*Pikh*+*Pi3/5/i*+*Pita*, *Pikh*+*Pita*+*Pi-zt*) identified in seedling blast tests showed a panicle blast average disease grade of lower than 4; in fact, these two combinations had also been identified in panicle blast tests. Therefore, we reason that elite *R*-gene combinations identified in panicle blast may also have high breeding potentials for improving seedling blast resistance (Figure 4C,E). 

### 2.5. Distribution Frequency of the Core R-Gene Combinations in Jiangsu Geng Rice Varieties 

Comparatively, the *Pita*+*Pia* combination predominantly existed in elite *R*-gene combinations identified in panicle blast tests (Figure 3B). In particular, this gene combination showed synergistic effects with other elite *R*-gene combinations (Figure 3C). In addition, the *Pita*+*Pi3/5/i* combination was detected in four of five MCS inoculations (Figure 3B). Notably, the varieties with these two dual *R*-gene combinations also displayed high resistance to seedling blast (Figure 4D). Therefore, we found that *Pita*+*Pia* and *Pita*+*Pi3/5/i* stood out as the most prominent elite core *R*-gene combinations, and that only 11 varieties contained *Pita*+*Pi3/5/i* and 30 varieties contained *Pita*+*Pia*, accounting for 5.8% and 15.8% of the total tested varieties, respectively (Appendix A). Although these varieties were from around 20 different breeding institutes, they were more concentrated in a few agricultural institutes, such as the Huaian Agricultural Research Institute and Wujin Rice Research Institute (Appendix A). 

Jiangsu *Geng* rice varieties could be divided into four ecotypes, namely, Zhong-Shu Zhong-Geng (ZSZG), Chi-Shu Zhong-Geng (CSZG), Zao-Shu Wan-Geng (ZSWG), and Zhong-Shu Wan-Geng (ZhSWG), on the basis of their heading dates. We further compared the distribution frequency of the two core *R*-gene combinations in these ecotypes. The results showed that ZSZG varieties had the highest frequency, followed by CSZG and ZSWG, while no apparent differences were observed among varieties of the three ecotypes, especially between ZSZG and CSZG varieties (Figure 5A). No ZhSWG varieties contained either of the two core *R*-gene combinations, which was probably due to the relatively small samples of ZhSWG varieties used here. We also compared the disease grades of the different ecotypic varieties with and without the core *R*-gene combinations. The results showed that the varieties of the three ecotypes containing the core *R*-gene combinations all showed a significant reduction on disease grades compared with controls. No significant differences were found among four ecotypes without the core *R*-gene combinations or among the three ecotypes with the core *R*-gene combinations (Figure 5A). In the 68 varieties tested for seedling blast, no ZhSWG varieties were found among them, and CSZG varieties had the highest frequency of the two core *R*-gene combinations. No significant differences were found on RF among varieties of the three ecotypes with the core *R*-gene combinations or among those without the core R-gene combinations (Appendix A). 

Because the varieties tested in this study were collected from different years, we further analyzed the distribution frequency of the two *R*-gene combinations across years, except for the varieties in 2001–2010 due to small samples. As shown in Figure 5B, the distribution frequencies of *Pita* and *Pia* clearly showed an upward trend with time, especially for *Pita*, which reached 67.5% in 2020–2022, clearly higher than 35.7% in 2011–2013. As a result, the *Pita*+*Pia* combination was also increased, reaching 15% in 2020–2022 compared to 0% in 2011–2013. On the contrary, the frequency of *Pi3/5/i* was not significantly increased in the past 10 years, resulting in the low frequency of *Pita*+*Pi3/5/i* (Figure 5B). 

According to the relatively low distribution frequencies of *Pia* and *Pi3/5/i* and the high frequency of *Pita*, varieties simultaneously containing *Pia* and *Pi3/5/i* are expected to be rare. In fact, only two varieties (Huaidao 182 and Lianjiajing 1) contained both *Pia* and *Pi3/5/i* genes in this study, which hindered the evaluation of its resistant effects. Through marker-assisted selection, we previously bred a new variety, Wankennuo 3 (WKN3), that simultaneously contained *Pia* and *Pi3/5/i* as well as *Pib* (Figure 5C). We evaluated the resistance performance of WKN3 at three different field locations across three years (2019–2021) in Anhui Province. Compared with the control variety (WuyunGeng 23) containing *Pikh* and *Pib*, we found that WKN3 showed clearly higher resistance to panicle blast at all locations and in different years (Figure 5D), suggesting a high breeding potential for the *Pia*+*Pi3/5/i* combination against panicle blast. Considering that more than a half *Geng* rice varieties already contain *Pita* in Jiangsu, varieties WKN3, Huaidao 182, and Lianjiajing 1 should serve as excellent doners to simultaneously provide *Pia* and *Pi3/5/i* for the development of new varieties with both core *R*-gene combinations containing three *R* genes (*Pita*+*Pia*+*Pi3/5/i*). 

## 3. Discussion

### 3.1. Identification of Elite R Genes and R-Gene Combinations against Panicle Blast in Jiangsu Geng Rice Varieties 

Among the blast diseases that develop on different parts of the host, panicle blast is the most serious one because it directly causes grain yield loss [3]. However, due to the difficulty in panicle blast inoculation, identification of *R* genes with durable, broad-spectrum resistance to panicle blast is difficult. In fact, almost all *R* genes were isolated through inoculation and phenotypic evaluation at seedling stage, and some of the *R* genes were found not conferring resistance to panicle blast [11]. In this study, we found that although the seedling blast resistance spectrum presents a significant positive correlation with panicle blast resistance, the *R*^2^ value (0.3097) was not very high (Figure 4A). Some cultivars showed resistance to more than 80% isolates when tested at seedling stage but displayed a susceptible phenotype in panicle inoculations (Appendix A). These results suggest that seedling blast resistance cannot completely represent panicle blast resistance. Regarding genes, we found that *Pi-zt* and *Pikh* showed significant contributions to seedling blast resistance but appeared only once in panicle blast resistance (Figure 2C and Figure 3A). Some *R* genes, such as *Pi9* and *Pigm*, have been found showing good resistance to both seedling blast and panicle blast, displaying a great potential for rice breeding [1]. Unfortunately, most *Geng* rice cultivars in the Jiangsu area do not contain either of these two elite genes (Figure 2A). Through MAS, we previously developed two *Geng* rice lines harboring *Pigm* that both significantly broadened their resistance spectrum at seedling stage to more than 184 isolates collected from most regions in the Yangtze River area. These two lines also showed excellent panicle blast resistance [15]. Due to its high performance, almost all breeding institutes in Jiangsu Province began to introduce *Pigm* into their breeding programs. However, due to the fact that arms race or coevolution exists between rice host and the blast pathogen, the wide cultivation of cultivars with a single *R* gene will undoubtedly lead to the change in pathogen populations or the rise of new pathogenic isolates, which will ultimately result in loss of resistance. Therefore, identification and utilization of more *R* genes conferring panicle blast resistance is of great importance to reduce rice yield loss. 

Compared with the enthusiasm of researchers on finding new blast *R* gene, studies on how to efficiently and reasonably use known *R* genes in rice breeding programs are rarely reported. Due to complex interactions between *R* genes and their genetic backgrounds, pyramiding of different genes may have different effects on resistance [1]. Through the development of near isogenic lines (NILs) with different *R* gene combinations, Wu et al. (2019) found that *Pigm*+*Pi*, *Pigm*+*Pi54,* and *Pigm*+*Pi33* displayed stable broad-spectrum resistance in artificial inoculations at both seedling and booting stages under different disease nurseries [1]. Therefore, identification of elite *R*-gene combinations that enhance panicle blast resistance is particularly important to breeders. NILs can be used to detect the interactions among *R* genes but cannot be used to study interactions related to genetic backgrounds. In fact, different cultivars in practice cannot share the same or similar genetic backgrounds. Therefore, the way in which to identify *R*-gene combinations with stable pyramiding effects in different cultivars or complex genetic backgrounds has been one of the critical issues that limit the use of these isolated *R* genes by breeders in breeding programs. 

Previously, we genotyped 158 rice cultivars using functional markers of 14 known *R* genes and evaluated their resistance levels to panicle blast through artificial inoculation with a group of mixed isolates [14]. Here, we genotyped 190 cultivars using the same 14 functional markers plus four additional markers corresponding to four other alleles in the *Pi2/9* locus and evaluated their resistance levels to five MCSs. The resistance levels to different MCSs showed positive correlations among them, but their correlation coefficients were not high, suggesting the diversity of isolates used in this study (Figure 1D). After the multinomial logistics regression analysis, we found that although a total of 11 *R* genes were detected that showed significant contributions to panicle blast resistance, and only 6 of them were found to confer resistance to at least two MCSs (Figure 3A). *Pita* and *Pi3/5/i* showed significant contributions to all five MCSs, and *Pit* and *Pia* showed resistance to four and three groups, respectively, demonstrating the importance of these four genes on panicle blast resistance of *Geng* rice in Jiangsu area. For resistance effects, we found that gene combinations generally had more significant effects than individual genes. We identified *Pita*+*Pia* and *Pita*+*Pi3/5/i* as core *R*-gene combinations because cultivars contained one of them generally displayed high resistance to both panicle blast and seedling blast (Figure 2D and Figure 3C), which is consistent with our previous finding [14]. This indicates that these two *R*-gene combinations have relatively stable pyramiding effects in complex genetic backgrounds, which is of great importance to breeders. Unlike *Pigm* and *Pi9*, the two dual *R*-gene combinations did not show complete resistance, which allowed us to conclude that they are more important in developing durably resistant cultivars to panicle blast, at least in the Jiangsu area. 

Among different *R* genes for *R*-gene combinations, we found that *Pita* is required for most combinations. Notably, *Pita* individually did not confer resistance, but it could significantly enhance the resistance contributions of *Pia* or *Pi3/5/i* when pyramided together. This implies that unknown synergistic interaction effects, rather than additive effects, may exist between *Pita* and *Pia* or *Pi3/5/i*. Certainly, we also found a few cultivars that were susceptible or highly susceptible, even though they contained these dual *R*-gene combinations (Figure 3C). We think this is probably attributable to possible variations in the *Pita*, *Pia*, or *Pi3/5/i* gene itself in these cultivars, resulting in the loss of interaction. Another possibility is that a common factor required for the signaling of *Pita* and *Pia* or *Pi3/5/i* exists in most rice cultivars but has mutated in these susceptible cultivars. It will be very interesting to investigate the interaction mechanism between *Pita* and *Pia* or *Pi3/5/i* in the future.

### 3.2. Breeding for Resistant Rice Cultivars against Blast in Jiangsu Area and Other Ecoregions 

In this study, we found that around 80% cultivars contained *Pib* and more than 50% cultivars contained *Pikh* and *Pita*, which is consistent with previous reports [14]. However, only *Pita* was detected with significant contributions to panicle blast resistance. In 2014, *Pib* and *Pikh* were considered as having good resistance to panicle blast [20] but they are not now, which implies that these two genes may have lost their resistance to recent major blast strains or isolates in the Jiangsu area after being widely used. Besides *Pita*, we found five *R* genes (*Pia*, *Pi3/5/i*, *Pit*, *Pid3,* and *Pik*) that showed significant contributions in resistance to at least two MCSs. The presence of the five *R* genes utilized in *Geng* rice cultivars was all less than 30% (Figure 2A). Thus, it is important to increase the utilization of these genes, as well as broad-spectrum resistance genes such as *Pigm* and *Pi9*, in breeding programs. Since *Pita* can significantly enhance the resistance of *Pia* and *Pi3/5/i*, we further compared proportions of *Pita*, *Pia*, *Pi3/5/i*, *Pita*+*Pia*, and *Pita*+*Pi3/5/i* in cultivars collected from different years (Figure 5A). We found that the proportions of *Pita* and *Pia* were clearly increased, leading to a slight increase in the *Pita*+*Pia* combination, whereas the proportions of *Pi3/5/i* and *Pita*+*Pi3/5/i* remained almost unchanged across years. Due to relatively low proportions of *Pia* and *Pi3/5/i* in cultivars, the *Pia*+*Pi3/5/i* combination was seldom pyramided in practice. Considering that more than a half of cultivars contain *Pita* but very few cultivars contain *Pia*+*Pi3/5/i*, varieties such as WKN3 should serve as ideal parents in future breeding programs for breeding new rice lines with *Pita*+*Pia*, *Pita*+*Pi3/5/I*, or even *Pita*+*Pia*+*Pi3/5/i*. In addition, we also found some cultivars containing *Pita*+*Pia* or *Pita*+*Pi3/5/i* that could serve breeders’ need to select for appropriate parents in *Geng* rice breeding programs towards resistance against panicle blast (Appendix A). 

Previous studies found that pyramiding many *R* genes affected rice grain yield or agronomically important traits [1,12]. However, according to the analysis of grain yields for *Pita*+*Pia* (*Pi3/5/i*) in 190 varieties (Appendix A), we believe that the core *R*-gene combinations identified in our study will not significantly affect rice grain yield or agronomically important traits, even in various genetic backgrounds, at least for *Geng* rice varieties in Jiangsu Province. This is because the varieties used in the current study were in fact from either commercial cultivars or the latest lines authorized (on the way to commercialization) by the Jiangsu government. These varieties all have excellent performance on grain yield and other agronomical traits, indicating that the *R* genes pyramided in these varieties maintain a good balance with development and grain yield and likely have no negative effects on grain yield. Moreover, approximately 20% of commercial varieties with different genetic backgrounds already contain the *Pita*+*Pia* or *Pita*+*Pi3/5/i* core *R*-gene combination, further suggesting no negative effects from these two dual gene combinations on agronomically important traits including grain yield. Thus, these core *R*-gene combinations should be readily applied through MAS to enhance *Geng* rice resistance to the blast disease. 

Given that the *M. oryzae* populations in different rice planting regions probably have different isolates, the elite *R*-gene combinations identified in this study may not be appropriate for other ecoregions. To find appropriate *R* genes or *R*-gene combinations for a given region, the critical step is to collect representative blast isolates in that region and evaluate enough number of cultivars for resistance to these isolates. Once the phenotype is obtained, the subsequent genotyping and statistical analysis could follow the methods described here. In fact, besides individual broad-spectrum *R* genes, it is necessary to identify appropriate *R*-gene combinations for different ecoregions, which should be one of the important strategies towards breeding cultivars with durable blast resistance. 

## 4. Materials and Methods

### 4.1. Plant Materials and M. oryzae Isolates 

A total of 190 *Geng* rice cultivars/lines, mainly planted in Jiangsu province during 2001–2022, were used in this study. Among them, 115 cultivars had been authorized by either Jiangsu province or the Ministry of Agriculture of China, and the remaining 75 lines were selected from regional tests of new rice cultivars organized by Jiangsu Province in 2020, designated as new rice lines (Appendix A). 

A total of 58 *M. oryzae* isolates were used in seedling blast inoculations by the artificial spraying method. These isolates were firstly selected on the basis of the cluster information of 321 isolates using the genotyping data of 10 characterized *avirulence* genes in clustering analysis (Appendix A); then, the geographic locations of these isolates in each cluster were considered. Lastly, these isolates belonged to 4 clusters and 11 cities (Lianyungang, Xuzhou, Suqian, Huaian, Yancheng, Yangzhou, Taizhou, Nanjing, Zhenjiang, Nantong, and Changzhou) in Jiangsu Province (Appendix A). 

### 4.2. Conidial Suspensions 

Five groups of MCSs were used in panicle blast inoculations by artificial injection at rice booting stage. MCS1 was introduced from Yongfeng Liu’s lab of the Institute of Plant Protection, Jiangsu Academy of Agricultural Science, that was composed of 5 isolates, nos. 2020-110, 2020-183, 2020-321, 2020-113, and 2020-353, and was designated by Jiangsu Province for the evaluation of the resistant levels of new rice lines to panicle blast in 2020. The other 4 MCSs contained 24 isolates with 6 isolates per MCS. These 24 isolates were selected from the above 58 isolates used in seedling blast inoculation assays (Appendix A). 

### 4.3. DNA Extraction and Molecular Marker Analysis 

Genomic DNA samples of two-week-old rice leaves and *M. oryzae* mycelia were extracted using the CTAB method [21]. The functional or diagnostic markers listed in Appendix A were used to detect the genotypes of 18 cloned blast *R* genes (*Pib*, *Pikh*, *Pita*, *Pid3*, *Pid2*, *Pia*, *Pik*, *Pi-zt*, *Pb1*, *Pikm*, *Pi3/5/I*, *Pi1*, *Pit*, *Pi25*, *Pigm*, *Piz*, *Pi9*, *Pi2*) [17,22,23,24,25,26,27,28,29,30,31,32,33,34,35,36] in rice and 10 isolated *avirulence* genes (*Avr1-Co39*, *Avri-Pita*, *ACE1*, *Avr-Pib*, *PWL1*, *Avr-Pia*, *Avr-Pi9*, *Avr-Pizt*, *Avr-Pik*, and *Avr-Pii*) in *M. oryzae* strains [37,38,39,40,41]. PCR products were separated by electrophoresis on 4% agarose gels or 8% non-deformable polyacrylamide gels. 

### 4.4. Evaluation of Seedling Blast Resistance 

Rice seedling blast resistance was evaluated with the method described by Wu et al. (2015) [5]. All rice seeds were sown in plastic nursery trays and raised in an artificial climate chamber (27–30 °C, normal light). Each line contained 20 plants. At the 3–4 leaf stage, all rice seedlings were transferred into glass inoculation boxes for spraying inoculation. The cultivation of conidial isolates and inoculum preparation were performed according to Puri et al. (2008) [42]. The conidial suspensions of each isolate were prepared to a concentration of 5 × 10^4^ conidia/mL in 0.02% Tween-20 (as surfactant). Each conidial suspension was sprayed onto the leaf surface through the round hole on the side of the inoculation box with a vacuum pressure pump. Each nursery tray was inoculated with 40 mL of conidial suspension. After inoculation, the rice seedlings were kept in dark for 24 h (H), then transferred to transparent crates covered with cling films to maintain humidity. The transparent crates were placed in a growth chamber set at 25–28 °C for 7 days. Disease responses were recorded using a 0–9 rating scale reported by He et al. (2020) [43]. Rice cultivar Lijiangxintuanheigu (LTH) was used as a susceptible control, and all rice lines were tested three times with a completely randomized design. 

### 4.5. Evaluation of Panicle Blast Resistance 

The artificial identification of panicle blast resistance in the field was performed as described previously [2] using the mixed conidial suspensions (MCSs) containing five or six *M. oryzae* isolates. Isolates were mixed in equal proportions to form a conidial suspension of 30–40 conidia per field of view under a 10 × 10 microscope. All rice cultivars/lines were planted in Xinminzhou Lingang Industrial Park, Jingkou District, Zhenjiang City, Jiangsu Province, China. Each line contained 6 rows with 10 plants per row and a row spacing of 11.1 × 30 cm in a randomized complete block design. Artificial inoculation was carried out at the booting stage, and 5 tillers each line were injected with 1 mL conidial suspension of each MCS. Disease scores were rated at rice ripening stage according to the Technical Specification for Identification and Evaluation of Blast Resistance in Rice Variety Regional Test (NY/T 2646-2014) issued by the Ministry of Agriculture and Rural Affairs of the People’s Republic of China. According to their disease scores, each rice line was then classified into different resistant levels: highly resistant (0 ≤ disease grade < 1), resistant (1.0 ≤ disease grade < 3), moderately resistant (3.0 ≤ disease grade < 5.0), moderately susceptible (5.0 ≤ disease grade < 7.0), susceptible (7.0 ≤ disease grade < 9.0), and highly susceptible (disease grade = 9.0). On the basis of the disease scores of each panicle, the final disease grade of the variety was determined as follows: (1) If the disease grades of 5 inoculated panicles ranged less than 2, then the highest disease grade of the 5 inoculated panicles was used as the final disease grade of the variety. (2) If the 5 disease grades ranged more than 3 and the panicles with relatively high disease grades constituted the majority, then the highest disease grade of the 5 inoculated panicles was designated as the final disease grade of the variety. (3) If the 5 disease grades ranged more than 3 and the panicles with relatively low disease grades accounted for the majority, then the second disease grade of the 5 inoculated panicles from top to bottom was used as the final disease grade of the variety. 

### 4.6. Data Analysis

Excel 2020 and GraphPad Prism9.0 were used to organize and graph the various types of data obtained in the study. A correlation analysis between the number of *R* genes and the disease grades of these varieties was performed using the regression analysis model in the IBM SPSS Statistics 22. Analysis of disease grade differences among varieties with different *R* genes or *R*-gene combinations was performed using ANOVA and Student’s *t*-test in IBM SPSS Statistics 22. 

The logistic regression model in IBM SPSS Statistics 22 was used to analyze the contribution of different *R* genes to panicle blast resistance. The disease grades of the tested varieties were reclassified into two levels: The disease grades lower than or equal to 5 were defined as “1”, and the grades higher than 5 were defined as “2”. According to the results of multinomial logistics regression, we could obtain the regression coefficient (β), significance value (*P*), 95% confidence interval value (95% CI), and odds ratio (OR) between each *R* gene and the relevant rice blast disease grade. The results were displayed with forest plots.

For seedling blast analysis, the interactions between *R* genes and *M. oryzae* isolates were treated as a dimorphic trait and recorded as “0” when the phenotype was resistant and “1” when it was susceptible. Resistance levels of varieties were represented by resistance frequency (RF), which was calculated as follows: number of incompatible isolates/total number of isolates inoculated × 100%.

## Figures and Tables

**Figure 1 ijms-24-03984-f001:**
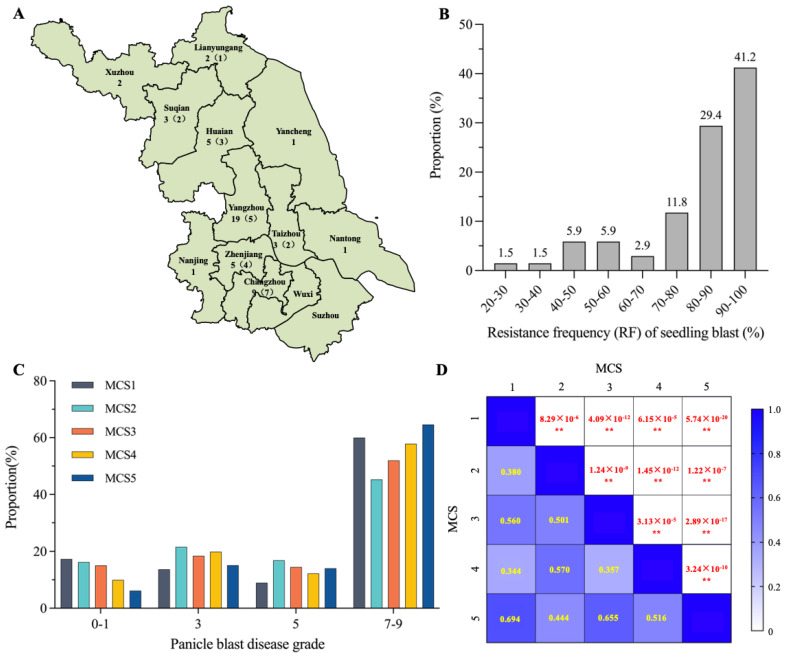
Evaluation of rice blast resistance of *Geng* rice varieties in the Jiangsu area. (**A**) Distribution of 58 *M. oryzae* isolates used for the artificial inoculation of rice blast in 11 prefectures of Jiangsu province. Numbers on the left denote the number of isolates used for seedling blast inoculation and numbers in parentheses on the right denote the number of isolates used for panicle blast inoculation. (**B**) Distribution of resistance frequency (RF) of seedling blast. The numbers above the columns denote the proportion of varieties with RF in the corresponding range. (**C**) Distribution of different panicle blast disease grades after inoculation with 5 groups of mixed conidial suspension (MCS). (**D**) Correlation of panicle blast disease grades with 5 groups of MCS. Yellow numbers denote the correlation coefficients of the 5 groups, and the red numbers denote the *p*-values. **: *p* < 0.01 by Student’s two-sided *t*-test.

**Figure 2 ijms-24-03984-f002:**
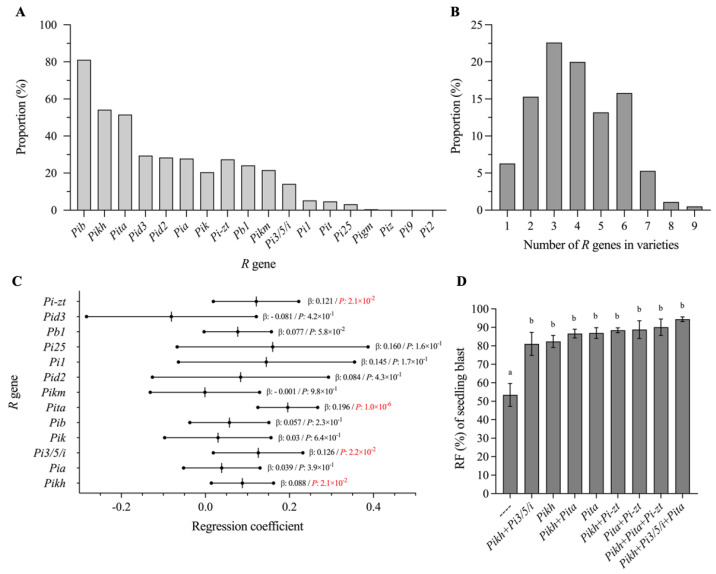
Distribution of 18 *R* genes in Jiangsu *Geng* rice varieties and analysis of elite *R* genes and *R*-gene combinations for seedling blast resistance. (**A**) Distribution frequency of 18 *R* genes in Jiangsu *Geng* rice varieties. (**B**) Distribution frequency of varieties that contain different numbers of *R* genes. (**C**) Multiple linear regression analysis of the contribution of each *R* gene and seedling blast resistance. “β” represents the regression coefficient. Red texts indicate that the contribution of a corresponding *R* gene to seedling blast resistance has reached a significant level (*p* < 0.05). (**D**) Correlation between RF and different elite *R* genes or *R*-gene combinations in seedling blast resistance. Different lowercase letters indicate significant differences according to *Duncan*’s multiple range tests (*p* < 0.05).

**Figure 3 ijms-24-03984-f003:**
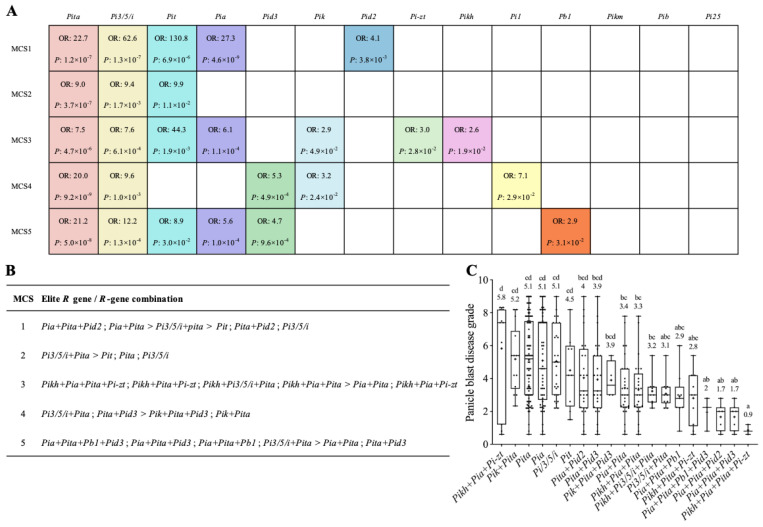
Analysis of elite *R* genes and R-gene combinations for panicle blast resistance. (**A**) Summary of the *R* genes that have significant contributions to panicle blast resistance in 5 MCS groups. OR represents the odds ratio. *p*-value compares varieties with and without the test *R* gene according to *Duncan*’s multiple range tests. (**B**) Summary of the elite *R* genes/*R*-gene combinations that have significant contributions to panicle blast resistance in each MCS inoculation. The panicle blast disease grades of those elite *R* genes/*R*-gene combinations were significantly lower than their corresponding control in respective MCS; their disease grades are in ascending order from left to right. “>” indicates significant differences according to *Duncan’s* multiple range tests. (**C**) Comparison of the panicle blast disease grades of different elite *R*-gene combinations. The numbers indicate the average disease grades. Different lowercase letters denote significant differences according to *Duncan*’s multiple range tests (*p* < 0.05).

**Figure 4 ijms-24-03984-f004:**
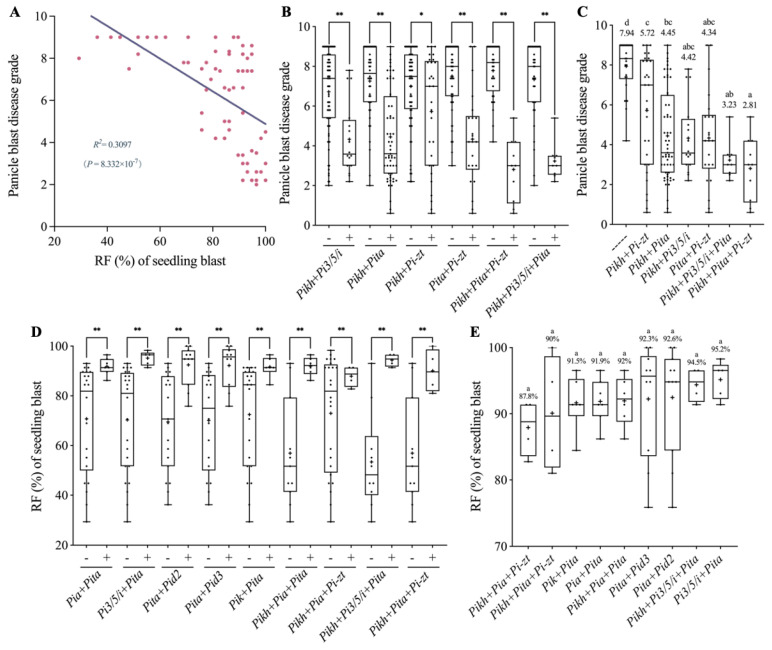
Correlation analysis between seedling blast resistance and panicle blast resistance. (**A**) Regression analysis of seedling blast RF and panicle blast disease grade. (**B**) Comparison of the panicle blast disease grades in 5 MCSs between elite *R*-gene combinations and corresponding controls from seedling blast resistance tests. “−” and “+” denote the varieties with or without corresponding gene combinations. **: *p* < 0.01 and *: *p* < 0.05 by Student’s two-sided *t*-test. (**C**) Comparison of panicle blast disease grades between different elite *R*-gene combinations from (**B**). The numbers indicate average disease grades. Different lowercase letters indicate significant differences according to Duncan’s multiple range tests (*p* < 0.05). (**D**) Comparison of the seedling blast RFs between elite *R*-gene combinations and corresponding controls from panicle blast resistance tests. “−” and “+” denote the varieties with or without corresponding gene combinations. **: *p* < 0.01 by Student’s two-sided *t*-test. (**E**) Comparison of the seedling blast RFs between different elite gene combinations from (**D**). The numbers indicate average RFs.

**Figure 5 ijms-24-03984-f005:**
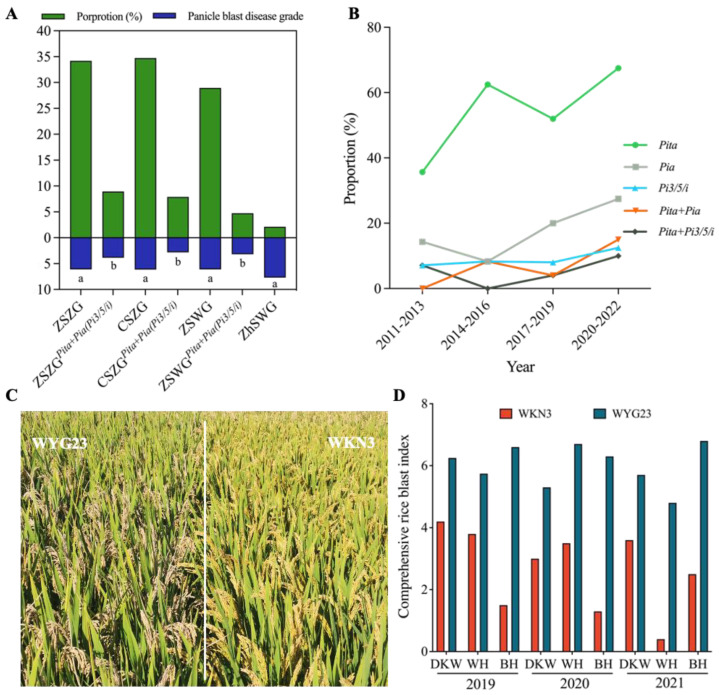
Application of core *R* genes and gene combinations in Jiangsu *Geng* varieties in recent years. (**A**) Application of *Pita*+*Pia* (*Pi3/5/i*) in varieties of different ecotypes tested for panicle blast resistance. Green columns denote the proportion of different ecotypes among the tested varieties, and blue columns represent the average panicle blast disease grades of different ecotypes. Different lowercase letters indicate significant differences according to *Duncan*’s multiple range tests (*p* < 0.05). (**B**) Distribution of 3 core *R* genes and 2 core *R*-gene combinations in Jiangsu *Geng* varieties at different annual intervals. (**C**) Panicle blast resistance performance of WNK3 in a blast nursery. WYG23 (WuyunGeng 23) was used as the control cultivar. (**D**) Comprehensive rice blast index for WKN3 and WYG23 at different blast nurseries in Anhui Province in 2019–2021. DKW: Dakuangwei; WH: Wanhe; BH: Baihu.

## Data Availability

The data presented in this study are available on request from the corresponding author.

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
