# Peer review of "Identification of Elite R-Gene Combinations against Blast Disease in Geng Rice Varieties"

_ijms, 2023, doi:10.3390/ijms24043984_

Round 1

Reviewer 1 Report

The manuscript is well written. Please define RF at the first instance (line no. 118). 

Author Response

  1. The manuscript is well written.Please define RF at the first instance (line no. 118). 

Response: We appreciate this valuable comment. In revised version, we have defined RF as resistance frequency at its first instance.

Reviewer 2 Report

This manuscript described evaluation for blast resistance in 190 Geng (japonica) rice cultivars collecting from Jiangsu area in China. The authors firstly evaluated seedling blast resistance with 58 isolates of Magnaporthe oryzae fungus. They revealed that two genes of Pikh and Pi3/5/i largely contribute to strong blast resistance in this cultivar set. Secondly, the authors evaluated panicle blast resistance with 24-mixed isolates of M. oryzae fungus. They revealed that gene combinations of Pita+Pi3/5/i and Pita +Pia showed strong resistance compared with other gene combinations. As the authors mentioned, the results in this study would provide valuable information for breeders to develop novel strong resistance cultivars. However, the authors should make several revisions in the current version of this manuscript described below.

1) At the Introduction and Discussion sections, the authors should explain and describe interaction effects between disease resistance and grain yield. Several previous studies have reported negative effects for pyramiding a lot of resistance genes to grain yield in various genetic backgrounds of rice cultivars. Generally, rice lines introducing more number of resistance genes show decreasing grain yield.

The authors should indicate data for grain yield in each 190 Geng rice cultivars (as Supplementary files), and should explain the data in this manuscript.

2) Figure 2D. there are two results for Pikh and Pi3/5/i+Pita here. I think the second box indicates the results for Pikh+Pi3/5/i.

3) Sentences on page 9 and Figure 5. The authors should indicate phenotyping data for seedling blast resistance in the seven ecotypes (described as Supplemental files). And the authors should indicate grain yields for these ecotypes.

4) This manuscript evaluated both seedling blast resistance and panicle blast resistance, and revealed significant gene combinations for each blast resistance. Therefore, you could change the manuscript title to 'Identification of elite R-gene combinations against panicle and seedling blast in Geng rice varieties'.

5) To indicate effectiveness of the appropriate gene combinations to blast fungus infection, the authors should carry out additional experiments including development and evaluation of near-isogenic lines and transformants introducing two blast resistance genes in the same genetic background.

Author Response

  1. At the Introduction and Discussion sections, the authors should explain and describe interaction effects between disease resistance and grain yield. Several previous studies have reported negative effects for pyramiding a lot of resistance genes to grain yield in various genetic backgrounds of rice cultivars. Generally, rice lines introducing more number of resistance genes show decreasing grain yield. The authors should indicate data for grain yield in each 190 Geng rice cultivars (as Supplementary files), and should explain the data in this manuscript.

Response: Thanks for comments. Yes, we agree with the reviewer that some negative effects from pyramiding many R genes on grain yield in various genetic backgrounds of rice cultivars may occur. In response to the comment, we have emphasized the importance in finding genes or R-gene combinations without inferior effects on grain yield in the Introduction and Discussion sections. In fact, our study does include this aim, but probably due to less-than-clear description, this purpose was not well perceived in our previous version. In this study, a total of 190 elite Geng rice cultivars/lines were used to analyze potential elite R-gene combinations against rice blast. All these varieties were from either commercial cultivars or the latest authorized lines (under the way for commercialization) in Jiangsu province in recent years. Therefore, all varieties were in fact having well performance on either grain yield or other importantly agronomical traits, which means that all different R genes pyramided in these varieties have been well-balanced with development and grain yield. Using these different varieties, some elite R-gene combinations (like Pita+Pia) identified in this study therefore will likely have no negative effects on grain yield. Around 20% varieties with different genetic backgrounds contained the elite R-gene combinations, Pita+Pia and Pita+Pi3/5/i, suggesting no negative effects from these two dual gene combinations on agronomically important traits including grain yield.

Certainly, according to the suggestions, we have added the grain yield data of all 190 varieties in “Supplementary Table 1”, and the analysis of the grain yields for Pita+Pia (Pi3/5/i) in all varieties in “Figure S7”.

  1. Figure 2D. there are two results for Pikh and Pi3/5/i+Pita here. I think the second box indicates the results for Pikh+Pi3/5/i.

Response: Thanks for pointing out our mistake. Yes, the second box in Figure2D indicates the results for Pikh+Pi3/5/i. We have corrected Figure 2D in revised manuscript.

  1. Sentences on page 9 and Figure 5. The authors should indicate phenotyping data for seedling blast resistance in the seven ecotypes (described as Supplemental files). And the authors should indicate grain yields for these ecotypes.

Response: Thanks for the comments. We have added the phenotyping data for the application of Pita+Pia (Pi3/5/i) in different ecotypes concerning seedling blast resistance and the results are shown in “Figure S6”. Also, the grain yield data for all these varieties with different ecotypes have been added to “Supplementary Table 1” and “Figure S7” in the revision.

  1. This manuscript evaluated both seedling blast resistance and panicle blast resistance, and revealed significant gene combinations for each blast resistance. Therefore, you could change the manuscript title to 'Identification of elite R-gene combinations against panicle and seedling blast in Geng rice varieties'.

Response: Thanks for the suggestion. Considering the fact that not all varieties were tested in the seedling blast evaluation assay. Besides, because blast disease actually includes panicle blast and seedling blast, we think that it is more appropriate to change the manuscript title to “Identification of elite R-gene combinations against blast disease in Geng rice varieties”.

  1. To indicate effectiveness of the appropriate gene combinations to blast fungus infection, the authors should carry out additional experiments including development and evaluation of near-isogenic lines and transformants introducing two blast resistance genes in the same genetic background.

Response: Thanks for the suggestion. We agree that these are good experiments and have planned to conduct in the future. However, we believe these experiments are beyond the scope of this manuscript because they require extensive efforts and are very time-consuming.

Reviewer 3 Report

Dear author

The manuscript is very important in the field

But need improvement in language editing

Abstract need improved

in results

Please choose the correct statistical test

how Used ANOVA or t-test although your data is nonparametric

please repeat all statistical test again with the correct test

For eg at fig 2D, you used the Duncan test; how is this data parametric? Please use the correct statistical test

at fig 3 a, what is the mean of color

In M&M

write again part statistical analysis with the correct test

where the conclusion

Author Response

  1. The manuscript is very important in the field. But need improvement in language editing.

Response: During revision, we have had the manuscript further improved by an experienced professional manuscript editor.

  1. Abstract need improved

Response: The abstract in revised manuscript has been carefully strengthened.

  1. in results, Please choose the correct statistical test, how Used ANOVA or t-test although your data is nonparametric, please repeat all statistical test again with the correct test. For eg at fig 2D, you used the Duncan test; how is this data parametric? Please use the correct statistical test.

Response: Thanks for your professional review of the manuscript. We have carefully reviewed our statistical methods and found no problems. Here, we would like to take "Fig. 2D" as an example to explain what we did. We firstly obtained 9 groups of varieties based on their genotypes on different gene loci, and employed the Duncan test in ANOVA to compare the resistance differences among the 9 groups. Each group contained different number of varieties, whose resistance values, namely RF or disease grades, were used as the original values in ANOVA. So, "Fig.2D" indicates that 8 groups of varieties with different R gene combinations all have significantly higher RF than the group without the 4 R genes, showing that significant contributions were detected in seedling blast assay listed in "Fig.2C". Similar statistical tests were used in the other analyses. This statistical method has also been used in several of our previous publications (Feng et al., Front. Plant Sci. 2022, 13:937767. doi: 10.3389/fpls.2022.937767; Wang et al., Chin. J. Rice Sci. 2020, 34, 413-424, doi:10.16819/j.1001-7216.2020.0208. in Chinese). Therefore, we believe the statistical methods used in this study are appropriate.

  1. at fig 3 a, what is the mean of color. In M&M.

Response: The different colors in this figure represent different R genes.

Reviewer 4 Report

I am interested in the results of the study entitled ‘Identification of elite R-gene combinations against panicle blast in Geng rice varieties.’ I inform you that it can be printed for the following reasons.

The information on panicle rupture provides useful information for developing Geng rice varieties. Blast caused by fungi is very difficult to control in rice. I believe that it will be very helpful for breeding disease-resistant varieties in the future because it seems that genetic considerations and experiments have been done well. However, what is a little disappointing is the lack of packaging materials.

Author Response

  1. I am interested in the results of the study entitled ‘Identification of elite R-gene combinations against panicle blast in Geng rice varieties.’ I inform you that it can be printed for the following reasons. The information on panicle rupture provides useful information for developing Geng rice varieties. Blast caused by fungi is very difficult to control in rice. I believe that it will be very helpful for breeding disease-resistant varieties in the future because it seems that genetic considerations and experiments have been done well. However, what is a little disappointing is the lack of packaging materials.

Response: Thanks for your recognition of the study. If by “packaging materials” you mean the delivery of varieties containing the identified elite R-gene combinations, we have the development of these varieties underway. However, because it takes extensive efforts and is very time-consuming, we believe it is beyond the scope of this manuscript.

Round 2

Reviewer 2 Report

This manuscript was clearly revised based on the previous comments.

Additional experimental data such as grain yields and additional discussions for relationship between R-gene combination and grain yield would be helpful understanding pyramiding effects of disease resistance genes in genetic backgrounds of the elite rice cultivars in further rice breeding programs.

And, I also believe that your revised title accurately indicates the content of this manuscript.